# Caerin 1.1/1.9 Enhances Antitumour Immunity by Activating the IFN-α Response Signalling Pathway of Tumour Macrophages

**DOI:** 10.3390/cancers14235785

**Published:** 2022-11-24

**Authors:** Xiaodan Yang, Xiaosong Liu, Junjie Li, Pingping Zhang, Hejie Li, Guoqiang Chen, Wei Zhang, Tianfang Wang, Ian Frazer, Guoying Ni

**Affiliations:** 1The First Affiliated Hospital, Clinical Medical School, Guangdong Pharmaceutical University, Guangzhou 510080, China; 2Cancer Research Institute, First People’s Hospital of Foshan, Foshan 528000, China; 3School of Science, Engineering and Technology, University of the Sunshine Coast, Maroochydore BC, QLD 4558, Australia; 4Centre for Bioinnovation, University of the Sunshine Coast, Maroochydore BC, QLD 4558, Australia; 5Faculty of Medicine, University of Queensland Diamantina Institute, Translational Research Institute, the University of Queensland, Woolloongabba, QLD 4102, Australia

**Keywords:** caerin peptide, therapeutic vaccine, macrophage, IFN-α response signalling pathway

## Abstract

**Simple Summary:**

Cancer immunotherapy has been enabled by immune checkpoint blockade, but is ineffective in many patients, mainly due to the high complexity of the immune-suppressive tumour microenvironment (TME). Here, with a murine transplantable tumour model, we demonstrate that host-defence caerin 1.1/1.9 peptides can alter the macrophages in the TME from immunosuppressive to immune active phenotypes, which appeared consistent with the macrophages observed at early-stage cervical cancer patients. The modulated macrophages by caerin 1.1/1.9 show significant activation of Type I interferons response and *Stat1* signalling, rendering animals resistant to further tumour challenges. Our results indicate that caerin 1.1 and 1.9 treatment may enhance the efficacy of current immunotherapy modalities.

**Abstract:**

Macrophages are one of the essential components of the tumour microenvironment (TME) of many cancers and show complex heterogeneity and functions. More recent research has been focusing on the characterisation of tumour-associated macrophages (TAMs). Previously, our study demonstrated that caerin 1.1/1.9 peptides significantly improve the therapeutic efficacy of combined specific immunotherapy and immune checkpoint blockade in a murine transplantable tumour model (TC-1). In this study, the mice inoculated with TC-1 tumour were immunised differently. The TAMs were isolated using flow cytometry and characterised by cytokine ELISA. The survival rates of mice with different treatments containing caerin 1.1/19 were assessed comparatively, including those with/without macrophage depletion. The single-cell RNA sequencing (scRNA-seq) data of previous studies were integrated to further reveal the functions of TAMs with the treatments containing caerin 1.1/1.9. As a comparison, the TAMs of stage I and II cervical cancer patients were analysed using scRNA-seq analysis. We demonstrate that caerin induced tumour clearance is associated with infiltration of tumours by IL-12 secreting Ly6C+F4/80+ macrophages exhibiting enhanced IFN-α response signalling, renders animals resistant to further tumour challenge, which is lost after macrophage depletion. Our results indicate that caerin 1.1/1.9 treatment has great potential in improving current immunotherapy efficacy.

## 1. Introduction

Human papillomavirus (HPV) infection-related malignancy remains a severe public health problem worldwide, although HPV prophylactic vaccine has been introduced for more than a decade [1,2]. Worldwide, infection is responsible for approximately 15% of all new cancer cases annually, while HPV infection is the second most important cancer-causing infection, which accounted for 28.8% of all infection-related cancers in 2012 [3,4]. This proportion increased to 31.4% in 2018 [5]. Infections with HPV16 and 18 are responsible for 70% of cervical cancers (CC), which are the fourth most common cancer affecting women worldwide [6]. Moreover, head and neck epithelial carcinoma is increasingly attributed to infection with high-risk HPV, especially HPV16, in both developed and undeveloped countries [7]. Besides, virtually all HPV-attributable cancer in men is caused by infections with HPV16 and 18 [5]. HPV infection-related cancers are most severe in developing countries, where HPV prophylactic vaccination rates are low [8,9].

Clinically, the International Federation of Gynecology and Obstetrics (FIGO) Cervical cancer staging system is an effective prognostic factor in patients with cervical cancer and useful guidance for treatment [10]. Stage I CC cells have grown from the surface of the cervix into deeper tissues. Stage II CC (4 cm or larger) grows beyond the cervix and uterus but has not spread to the walls of the pelvis or the lower part of the vagina [11]. Surgery is often the treatment option followed by chemotherapy for these stages; however, the immunopathological profiles between different stages have been less studied.

Immunotherapy is becoming a routine modality for the management of cancers, following the introduction of immune checkpoint inhibitors and the transfer of CAR-T cells for cancer management [12]. However, the efficacy of immune checkpoint inhibitor monotherapy, such as the blockade of PD-1 [13] and CTLA-4 [14], is relatively low in the management of advanced CC [15]. Therapeutic vaccines aiming at eliciting tumour-specific T cells have been shown partially effective in CIN3 and VIN3 pre-cancer, but are ineffective against advanced cervical cancers [16,17].

In both prophylactical and therapeutic settings, therapeutic vaccination with the simultaneous blocking of interleukin 10 (IL-10) increased antigen-specific CD8+ T cell responses, which consequently improved tumour growth inhibition [18]. The efficacy of this therapy strategy is superior to that of the same vaccination without IL-10 signalling blockade [19]. PD-1 blockade combined with therapeutic vaccines are synergistic in animal models and clinical trials [20,21,22]. The efficacy of PD-1 blockade and therapeutic vaccination becomes more significant after intra-tumoral injection of host-defence caerin 1.1/1.9 peptides [23].

Caerin 1.1/1.9 peptides are able to inhibit the proliferation of TC-1 [24,25], HeLa [26] and A549 [27] cells in vitro, and induce cell apoptosis, probably through the stimulation of the TNF-α signalling pathway [26,28]. Caerin 1.1/1.9 inhibited TC-1 tumour growth and modified the functions of tumour infiltrating immune cells, such as T cells, NK cells, and dendritic cells in vivo [23,29], and more activated CD8+ T cells and NK cells were recruited to the tumour sites [23,28,29]. Caerin 1.1/1.9 also modulated macrophage heterogeneity within the tumour, drastically reducing the M2 type while increasing the M1 type macrophages [23,29].

Macrophages are one of the essential components of the tumour microenvironment (TME) [30]. The high density of certain tumour-associated macrophage (TAM) phenotypes in the TME is often associated with key processes in tumour progression, such as angiogenesis, immunosuppression, metastasis, and/or a poor prognosis [31,32,33]. TAMs can display opposing phenotypes and functions that are either tumoricidal, M1-like, or immune-suppressive/tumour-supportive, M2-like [34]. M1-like MΦs are activated by toll-like receptor (TLR) agonists and Th1 cytokines, such as interferon-gamma (IFN-γ), which empowers them with the ability to kill and phagocytose pathogens. These MΦs normally show upregulated proinflammatory cytokines such as interleukin (IL)-1β and IL-12, as well as the elevation of tumour necrosis factor-α (TNF-α) and reactive molecular species. M1-like MΦs present antigens via major histocompatibility complex (MHC) class II molecules. Th2 cytokines, like IL-4 and 13, stimulate monocytes/macrophages to express an M2 activation state. M2-like MΦs produce high levels of anti-inflammatory cytokines, such as IL-10, which amplifies metabolic pathways that can suppress adaptive immune responses [35]. Single-cell RNA sequencing (scRNA-seq) found that TAM is composed of heterogenous populations of MΦs with different biological functions. Targeting macrophages achieves better or no antitumour responses [30,36,37,38,39,40], indicating the functional status of the tumour macrophage is of great importance for the outcome of immunotherapy.

In this work, we revisited two scRNA-seq studies on the intratumoural injection [23] and topical application [29] of caerin 1.1/1.9 in treating TC-1 tumour in mice, to reveal the signalling pathway(s) regulated by caerin 1.1/1.9 in different macrophage phenotypes. We demonstrated that caerin 1.1/1.9 significantly activated the IFN-α response pathway of multiple macrophage types present in the TME. The reprogramming of M2 to M1-like phenotypes was induced by caerin 1.1/1.9, which became the dominant population in the TME and significantly increased the efficacy of the specific immunotherapy developed in this work. In addition, we compared the heterogeneity and functions of macrophages identified from the TME of cervical cancer (CC) stage I (AIII, CIN3) and stage II (IIB) patients using scRNA-seq analysis. The macrophage phenotypes were characterised, with those derived from the stage I patients showing significant activation of the IFN-α response pathway. These results pinpointed comparative immunological footprints of macrophages between the treatments containing caerin 1.1/1.9 in murine models and human CC stage I patients, strongly suggesting that caerin 1.1/1.9 are promising agents to be included in cancer immunotherapy that could alter the TME to be more immune active.

## 2. Materials and Methods

### 2.1. Cervical Cancer Patient Information

Seven patients with pathological confirmed cervical cancer who underwent surgical removal of cervical cancer at the Department of Obstetrics and Gynecology at Foshan First People’s Hospital, between 2019 and 2021 were included in the current study, with three at AIII, one at CIN3 (referred to as CCI) and three patients at IIB (referred as CCII). The ethical approval code was L2016 (13).

### 2.2. Mice

Experiments were approved by Animal Experimentation Ethics Committee (Ethics Approval Number: FAHGPU20160316). Specific pathogen-free (SPF) six to eight weeks old adult female C57BL/6 (H-2b) mice were ordered from the Animal Resource Centre of Guangdong Province China. Mice were kept under SPF conditions at the Animal Facility of the first affiliated hospital of Guangdong Pharmaceutical University. Five mice were kept in each cage at 22 °C and 75% humidity and provided with sterilised standard mouse food and water and a 12-h light/12-h dark cycle.

### 2.3. Cell Line, Peptide Synthesis and Antibodies

Transformed with HPV16 E6/E7, a murine TC-1 cell line was purchased from Shanghai Institute for Cell Resources Centre of Chinese Academy of Sciences. Cell culture was following the procedure described previously [28].

Caerin 1.1 (GLLSVLGSVAKHVLPHVVPVIAEHL-NH_2_), caerin 1.9 (GLFGVLGSIAKHVLPHVVPVIAEKL-NH_2_), and a control peptide P3 (GTELPSPPSVWFEAEFK-OH) were synthesised by Mimotopes Proprietary Limited (Wuxi, China), with a purity above 99% and the concentrations of lipopolysaccharide below 0.44 EU mL^−1^.

HPV16 E7 CTL epitope RAHYNIVTF-OH, and four overlapping peptides representing the entire HPV 16 E7 protein, EX (MHGDTPTLHEYMLDLQPETTDLYCYEQLNDSSEEE-OH, LNDSSEEEDEIDGPAGQAEPDRAHYNIVTFCCKC-OH, DRAHYNIVTFCCKCDSTLRLCVQSTHVDIR-OH, CVQSTHVDIRTLEDLLMGTLGIVCPICSQKP-OH were synthesised (purity > 99%) by Mimotopes Proprietary Limited (Wuxi, China).

Rat anti-mouse anti-IL10 receptor (1B1.3), Anti-PD1 (J43) Hamster anti-mouse monoclonal antibody and IgG Isotype control antibody (LTF-2) were purchased from BioXcell, USA. Rat Anti-Mouse fluorescent conjugated antibodies were purchased from BD, Biolegend and eBioscienc, including FITC-CD45.2 (clone: 104), PE-F4/80 (clone:BM8), APC-Ly6C (clone: HK1.4), PerCP-Cyanine5.5-CD11b (clone: M1/70), BV421-MHCII (clone: M5/114.15.2), PE-Cy7-CD11c (clone: HL3), PE-Cy7-CD86 (clone: GL1), APC-Cy7-CD3 (clone: 145-2C11), PE-Cy7-PD-L1 (clone: 10F.9G2), and Fixable Viability Stain 510 (FVS510). They were stored at −80 °C until further use.

### 2.4. Tumour Challenge

TC-1 cells, at approximately 70% confluency, were harvested with 0.25% trypsin and washed repeatedly with PBS. About 5 × 10^5^ cells/mouse in 0.2 mL of PBS were injected subcutaneously into the left flank. The average diameter of tumour size was assessed every 3 days using callipers, calculated as width × width × length. Mice were given 1% sodium pentobarbital by i.p. injection when treatment was performed. At the end of each experiment, mice were sacrificed when the tumour diameter reached 15 mm (for survival curve), by CO_2_ inhalation.

### 2.5. Immunisation of Mice

Three days post-TC-1 challenge, when tumours were palpable, mice were immunised intramuscularly (i.m.) with the vaccine on day 3, day 9 and 18, respectively. The vaccine contains 40 µg of four overlapping HPV16E7 peptides (EX) (10 µg of each peptide), 10 µg of monophosphoryl lipid A (MPLA) (Sigma-Aldrich Corp., St. Louis, MI, USA), with or without 300 µg of anti-IL10R antibodies, dissolved in 100 µL of PBS; 300 µg of anti-PD-1 was administered intraperitoneally (i.p.) on days 9 and 21 after the tumour challenge.

### 2.6. Tumour Local Administration of Caerin Peptide

Three days post-TC-1 challenge, the tumour diameters reached 3 to 5 mm and the mice were intratumourally injected with 30 µg of caerin peptides (caerin 1.1/1.9) in PBS daily for six consecutive days.

### 2.7. Depletion of Macrophage

Mice were injected intraperitoneally (i.p.) with 200 μL of Clodronate Liposomes (5 mg/mL, Clodronate Liposomes Organization, Amsterdam, The Netherlands) or Control Liposomes, and 24 h later injected subcutaneously (s.c.) with 5 × 105 TC-1 cells/mouse in 0.2 mL of PBS and into the right flank. Systemic and local macrophage depletion was achieved by injection of Clodronate Liposomes concomitantly intraperitoneally (200 μL) and subcutaneously near the tumour cell injection site (100 μL) per 4 days until the end of the experiment.

### 2.8. Flow Cytometry

To obtain single cells, TC-1 tumour tissues were isolated and homogenised with a tumour dissociation kit (Miltenyi Biotec, Bergisch Gladbach, Germany). Single cells were stained with FITC-CD45.2 (clone: 104), APC-Cy7-CD3 (clone: 145-2C11), PE-Cy7-CD8a (clone: 53–6.7), BV421- INFg (clone: XMG1.2), PE-B220 (clone: RA3-6B2) and FVS510. Viable cells were separated and analysed on a flow cytometer (FACS Aria II; BD Biosciences, San Jose, CA, USA). Data were analysed with Flow Jo v10.0 software (Tree Star Inc., Ashland, OR, USA).

### 2.9. Cytokine ELISA for IL-12, IL-10, IL-6 and TNF-α

Murine cytokine ELISA kits were purchased from BioLegend (San Diego, CA, USA), and were performed following the instructions of the manufacturer. 

### 2.10. Single Cell RNA Sequencing Data Analysis

Previously, we performed single-cell RNA sequencing of CD45+ TC-1 tumour infiltrating cells and quantitative analysis of TC-1 tumours of application of caerin 1.1/1.9 and combination therapy of caerin 1.1/1.9 with ICB and therapeutic vaccination [23,29]. The gene expression data were downloaded from https://singlecell.broadinstitute.org/single_cell (accessed on the 13 May 2022), the accession numbers were SCP1371 and SCP1093.

For the scRNA-seq analysis of CC patients, the tumour tissues were collected with consent from the Human Ethnic Committee of the Foshan First People’s Hospital (No: L2016). Cells were washed once with ice-cold PBS containing 10% foetal bovine serum post-sorting and counted using a haemocytometer. The cells were then loaded onto a 10× chromium machine (10× Genomics, San Francisco, CA, USA) and run through the library preparation procedures following guidance from the Chromium Single Cell 3′ Reagent Kits v2 (10× Genomics, Pleasanton, CA, USA). The data were analysed using the pipeline described elsewhere [23,29].

### 2.11. Protein-Protein Interaction (PPI) Analysis

Interactions among the proteins encoded by significantly regulated genes were predicted using STRING [41]. A required interaction score of 0.700 was used. Neither the first nor second shell of the PPI was included. Topological and statistical analyses were performed to explore the potential functions in our constructed network using the NetworkAnalyzer plugin in Cytoscape 3.7.1 [42]. Proteins without any interaction were excluded from the final network, which was visualised using Cytoscape.

### 2.12. GSEA Analysis

The genes differentially expressed between different treatments were analysed by Gene Set Enrichment Analysis (GSEA) with *p*-value < 0.05 using GSEA v4.1.0 [43], to predict the Hallmark pathways enriched and the ranking of the genes associated with each pathway.

### 2.13. SCENIC Analysis of Transcription Factor

The transcription factor network inference was analysed using the SCENIC R package [44]. Briefly, a log-normalised expression matrix generated using Seurat was used as input. The gene co-expression network was identified by GENIE3 [45]. Each module was pruned based on a regulatory motif near a transcription start site via RcisTarget. The networks were retained if the TF-binding motif was enriched among its targets. The target genes without direct TF-binding motifs were removed. The retained networks were considered regulons. The activity of each regulon for every single cell was scored via the AUC scores using the AUCell R package.

### 2.14. Real Time PCR

The total RNA of tumour tissues was isolated with TRIzol reagent (Thermo Scientific, Waltham, MA, USA). First-strand cDNA was synthesised from 1 μg total RNA using a HiScript^®^ II Q RT SuperMix for qPCR Kit (Vazyme, Nanjing, China). The ChamQ SYBR qPCR Master Mix (Vazyme, Nanjing, China) was used for StepOnePlus™ Real-Time PCR System with Tower (Applied Biosystems Inc, Foster City, CA, USA). The cycling conditions were: 90 s at 95 °C, 40 cycles at 95 °C for 5 s, 60 °C for 30 s and 72 °C for 20 s. β-actin (*Actb*) was used as an internal control. The relative level was calculated by the relative quantification 2-ΔΔCT method. The primer sequences were listed in Table 1.

## 3. Results

### 3.1. Immunotherapy plus Caerin 1.1/1.9 Induces Tumour Macrophages to Secrete More IL-12 and Less IL-6 and IL-10

Previously, we demonstrated that intratumoural injection with caerin 1.1/1.9 (caerin) significantly increases the survival time of TC-1 tumour bearing mice treated with HPV16 peptides/monophosphoryl lipid A/anti-IL-10 and anti-PD-1 therapy (Double-therapy, DT). DT plus caerin significantly reduced M2 type MΦs but increased pro-inflammatory macrophages in the TME [23]. Here, we examine the characteristics of the macrophages in a TC-1 tumour, following DT with caerin or a control P3 peptide (control). Tumours from untreated (UN) mice had more F4/80+Ly6C− macrophages (56.4%) and less F4/80+Ly6C+ (27.7%) macrophages and tumours from mice receiving DT and caerin had less F4/80+Ly6C− (34.19%) and more F4/80+Ly6C+ (55.33%) macrophages, whereas tumours from mice given DT with the control showed similar numbers of F4/80+Ly6C− (46.69%) and F4/80+Ly6C+ (45.59%) macrophages (Figure 1A–C and Appendix A). F4/80+Ly6C− macrophages from both treatment groups secreted less IL-12 and more IL-10, IL-6 and TNF-α than F4/80+Ly6C+ macrophages (Figure 1D and Appendix A). The highest levels of secreted IL-12 and lowest levels of secreted IL-10, IL-6 and TNF-α were produced by F4/80+Ly6C+ macrophages from the ‘Caerin + DT’ treated group.

### 3.2. Depletion of Macrophages Attenuate the Caerin 1.1/1.9 Enhanced Efficacy of Immunotherapy

To test the hypothesis that tumour macrophages are necessary for caerin-mediated anti-tumour responses, TC-1 tumour bearing mice treated with DT with caerin or control, and untreated tumour bearing mice, were treated with clodronate liposomes (CL) to deplete MΦs (Figure 2A). Clodronate liposomes reduced the number of tumour MΦs (CD45+F4/80+) by approximately 30% in untreated TC-1 tumour-bearing mice. Tumour MΦs in the ‘DT + caerin’ treated group were elevated by more than 200%, and in the DT treated group by 100%, when compared to the untreated group, and were reduced in each case by about 40% with the addition of CL (Figure 2B and Appendix A). Analysis of tumour infiltrating cells revealed that both the numbers of CD45+ and CD3+ T cells were reduced in the CL macrophage-depleted group when compared with macrophage-competent animals (Figure 2B and Appendix A; Appendix A). The addition of CL did not affect the tumour growth in the untreated group (Figure 2C and Appendix A). TC-1 tumour bearing mice treated with DT plus caerin had palpable tumours after 12 to 20 days which grew more slowly (Figure 2C and Appendix A) than mice receiving DT with control and had an increased survival rate (Figure 2D and Appendix A), whereas the shorter time to palpable tumours and increased survival with caerin treatment was not observed after CL treatment. The average survival time of mice receiving DT + caerin without CL macrophage depletion was 56 days, whereas those with CL macrophage depletion was 27 days. The survival time of untreated mice with or without CL macrophage depletion were similar (17 days), suggesting that macrophages contributed specifically to the therapeutic effect of caerin in TC-1 tumour-bearing mice. Moreover, a fraction of TC-1 tumour-bearing mice, ranging from 10 to 50%, completely cleared the tumour following DT + caerin (Figure 2D). Notably, mice that rejected the TC-1 tumour following DT + caerin were rechallenged with TC-1 tumour 50 days after initial tumour inoculation and did not grow palpable tumours (Figure 2E).

### 3.3. Caerin 1.1/1.9 Activates IFN-α Response Pathway in Tumour Macrophages

Aberrant expression of *Arg1* in the TME and in tumour macrophages has been found to correlate with tumour growth, metastasis, and poor prognosis in many types of cancers [46,47]. Previously, we showed that the topical application of a temperature-sensitive gel containing caerin 1.1 and 1.9 largely altered the functions of four *Arg1^hi^* macrophage populations (including Arg1B, Arg1A, *Ear2^hi^* and MΦ/DCs), to be more pro-inflammatory and immune response active, with respect to the untreated or control group [29]. The INF-α response Hallmark pathway was highly activated in all *Arg1^hi^* MΦs from the TC-1 tumour bearing mice treated with caerin. In the current study, the scRNA-seq data of this previous work, particularly the macrophages, were reanalysed.

Notably, the enrichment of this INF-α response pathway was also detected in the other MΦ subtypes of the caerin relative to the control group, including resident-like MΦs and TAMs (Figure 3A,B); however, this was not the case for M1 MΦs (Appendix A). The differentially expressed genes (DEGs) of at least one MΦ subtype associated with the activation of INF-α response were analysed (Appendix A), which presented 54 DEGs mutually identified in the six MΦ subtypes (Appendix A). The expression (in FC values) of the 54 DEGs, with respect to the control, were compared in six MΦ populations (Appendix A). Many genes were significantly overexpressed (FC ≥ 1.2) in the TAMs and *Ear2^hi^* MΦs from the caerin-treated animals when compared to TAMs from the control group, particularly *Herc6*, *Parp9*, *Ripk2* and *Ube2l6* (Appendix A). *Ifit3*, *Samd9l*, *Irf7*, *Stat2*, *Psmb9* and *Tap1* were upregulated in nearly all MΦ subtypes. The corresponding proteins are functionally linked to each other and to many other gene products involved in immune regulation (Appendix A). A comparison of the average expression of the DEGs in at least one MΦ subtype of caerin showed that *Cebpb* was upregulated in all populations except M1 MΦs (Appendix A). Significant upregulation of *Coro7* was detected in five MΦ populations, and its expression significantly correlated (*p* = 0.0044) with the survival of patients with CESC (Figure 3C). Similarly, the correlation between the upregulation of *Ccl4* (*p* = 0.041) and *Gcnt2* (*p* = 0.021) with CESC survival time was identified, respectively (Figure 3D,E). The qPCR results confirmed the upregulation of *Coro7* (*p* = 0.04), *Ccl4* (non-significant) and *Gcnt2* (*p* = 0.011) in the caerin treated tumour bearing mice when compared to untreated mice (Figure 3F). In addition, *Stat1* was significantly elevated; *Stat1* was previously identified, by both scRNA-seq and quantitative proteomic analysis, as one of the key genes involved in the signalling regulation induced by the caerin peptides [23,29].

### 3.4. Caerin plus DT Immunotherapy Activates the Transcription Factors Regulating Pro-Inflammatory Genes of Tumour Macrophages

To reveal the transcription factors (TFs) that might be regulating macrophage phenotypes, we examined the TF regulatory network of the six MΦ subtypes as mentioned above using SCENIC analysis. A total of 92 regulons were differentially detected among the MΦ subtypes (Appendix A). Upregulated interferon regulatory factors, the *Irf1*, *Irf2*, *Irf7*, and *Irf9* regulons, significantly distinguished M2 MΦs from the other MΦs, particularly the *MHCII^hi^* MΦs. Significant upregulation of the *Bmyc*, *Cebpg*, *Mx1* and *Maf* regulons was observed in the *MHCII^hi^* MΦ and TAMs, whereas expression of the genes regulated by *Cebpb*, *Nfil3*, *Jund*, *Fosl2*, *Jun*, and *Fos* was observed in M2 and *MHCII^hi^* MΦs.

We examined the impact of different tumour treatments on macrophage gene expression. Genes regulated by *Jun*, *Jund*, *Fos*, *Junb* and *Fosl2* were upregulated in the MΦs from the caerin-treated animals (Figure 4A) and were more associated with the M2, *MHCII^hi^* MΦs and MΦ/DCs (Appendix A). In addition, the SCENIC analysis showed that the genes regulated by the TFs, including *Maz*, *Runx1*, *Zmiz1*, *Fli1* and *Elk3*, were downregulated in *Ear2^hi^* MΦs. The GSEA analysis reveals that the *Ear2^hi^* MΦs were enriched with the activation of IL2/STAT5, TGFβ and TNF-α via NFκB signalling pathways (Figure 4B). The enriched DNA-binding motifs corresponding to the TFs regulated by the caerin treatment were comparatively displayed, and all of them show a high propensity for hydrophobicity (Appendix A). The top 18 immune response-relevant genes significantly expressed in at least one MΦ population in the caerin-treated animals when compared to untreated animals are displayed in Figure 4C. Notably, the pseudogene *Gm8797* was upregulated in all MΦ subtypes. Several genes associated with TNF-α signalling, including *Tnfrsf11a*, *Tnfaip3* and *Tnfaip2*, were elevated in TAMs, whereas *Clcf1* and *Il1b* were downregulated. These results prompt us to investigate whether the TNF-α pathway was the main signalling pathway that was upregulated in the tumour infiltrating macrophages, especially TAMs or *Arg^hi^* macrophages with immune suppressive characteristics.

### 3.5. Tumour Macrophages Isolated 7 Days after Tumour Inoculation from TC-1 Tumours Secreted more IL-12 than Those from Day 14

Comparing the population of tumour associated immune cells at 7 and 14 days after TC-1 tumour inoculation, we observed a significantly higher proportion of CD45+ cells at 7 days compared with 14 days (18.5% versus 11.0%: *p* < 0.0001) (Figure 5A and Appendix A). In addition, more MΦs were present at 7 than at 14 days (10.3% versus 6.30%; *p* < 0.001) (Figure 5B). Tumour MΦs were stained with CD45, CD11b, F4/80 and Ly6C and the CD45+D11b+ MΦs were thus separated into four major subpopulations (Figure 5C). Tumour resident F4/80+Ly6C^−^ and proinflammatory F4/80+Ly6C+ MΦs subpopulations accounted for over 80% of macrophages at day 7 and day 14. F4/80+Ly6C− and F4/80+Ly6C+ MΦs were sorted and stimulated with MPLA (100 ng/mL) overnight, and the supernatants were measured for IL-12, IL-10, IL-6 and TNF-α by ELISA (Figure 5D). IL-12 secretion by both F4/80+Ly6C− and F4/80+Ly6C+ cells was higher at 7 days than at 14 days. IL-10 and TNF-α secretion by F4/80+Ly6C+ macrophages was higher at 7 days, while F4/80+Ly6C− macrophages secreted more IL-6 at 14 days. High levels of IL-10 and IL-12 were secreted by F4/80+Ly6C+ MΦs at day 7, while IL-6 and TNF-α secretion was high from F4/80+Ly6C− MΦs at 7 and 14 days (Figure 5D).

### 3.6. IFN-α Response Signalling Pathway Activated Macrophages Are Increased in Early-Stage Cervical Cancer

The relative proportion of five macrophage phenotypes, including resident-like (C0-Res), TAM (C1-TAM), M2-like (C2-M2), MΦ/DCs (C3-DC), and M1-like (C4-M1) MΦs, was assessed for four patients with early (CCI) and three patients with late (CCII) stage cervical cancer, using scRNA-seq analysis (Figure 6). This was to test the hypothesis that if macrophage-mediated immunity controls early-stage tumours, there will be more macrophages expressing immune mediators in early than in late-stage cancer [48]. The MΦs from CCI patients generally showed a higher proportion of resident- and M2-like MΦs, and significantly lower TAMs, when compared with CCII patients (Figure 6A). Most differentially expressed interleukin and chemokine genes showed significantly higher expressions in MΦ populations from the CCI patients, especially in the M1-like MΦ (Appendix A). The genes upregulated in MΦ subtypes from tumours treated with topical caerin gel including *Stat2*, *Isg15*, *Tap1*, *Psmb9*, and *Parp9*, were more highly expressed in the CCI than CCII MΦs. IFN-α and IFN-γ signalling activation was observed in all subpopulations of MΦs in CCI subjects, and TNF-α signalling was observed in MΦ/DCs, resident- and M2-like MΦs (Figure 6B,C and Appendix A–E). IL6/JAK/STAT3 signalling was enhanced in M1-like MΦ of the CCI patient group, which was also activated in the mouse model in the resident-like MΦs, in TC1 tumours treated with the caerin peptide gel (Figure 3). We examined the DEGs between the CCI and CCII groups that were also significantly regulated by caerin in the murine models (Table 2). Notably, the genes overexpressed following the caerin treatment were also mostly overexpressed in M1- or M2-like macrophages in CCI when compared to the CCII subjects, suggesting that caerin 1.1a/1.9 could be used to induce these macrophage phenotypes to show increased expression of tumour controlling immune mediators.

## 4. Discussion

In this study, we demonstrated that the DT of ICB and therapeutic vaccination plus caerin 1.1/1.9 increased the proinflammatory cytokine IL12, while reducing IL10, IL6 and TNF-α secretion in tumour macrophages; the depletion of macrophages greatly attenuated the anti-tumour responses by DT boosted by caerin 1.1/1.9. By mining our previously published scRNA-seq data, we showed that tumour local administration (both topical application and intratumoural injection) of caerin 1.1/1.9 significantly activated the IFN-α response pathway in different phenotypes of tumour macrophages, in general, similar to that exhibited by the tumour macrophages of early CC patients. Caerin 1.1/1.9 also promoted the expression of several TFs playing key roles in the regulation of genes that result in a proinflammatory response.

Immunotherapy has become a routine modality for the treatment of cancers, especially with the introduction of ICB and CAR-T therapy, therapeutic vaccines in pre-clinical animal models have also demonstrated exciting efficacies. However, a great portion of cancer patients do not respond to ICB treatment. One of the most important caveats is that the immunosuppressive TME stunts the therapeutic effects of immunotherapy. Disturbing the TME is, therefore, a key issue to consider for immunotherapy. Given that caerin 1.1/1.9 can directly kill tumour cells and make the tumour cells release inflammatory cytokines we, therefore, set out to investigate whether caerin 1.1/1.9 can alter the TME and the efficacy of immunotherapy.

The enhanced efficacy of immunotherapy mediated by tumour local treatment with caerin peptides was mediated by macrophage, most likely through the IFNα signalling pathway as demonstrated by the scRNA-seq analysis of the tumour infiltrating macrophages. scRNA-seq analysis of macrophages between early and later-stage cervical cancer tumour samples also indicate that the IFNα signalling pathway was more activated in the early-stage cervical cancer.

Type I IFNs (IFN-α and IFN-β) constitute the first line of defence against both viruses and cancer cells, although it plays dual roles that can either prevent or promote tumour development [49]. Viruses have been found to use different mechanisms to disturb type I IFN signalling. HPV18E6 directly binds to *Tyk2* and impairs Jak-STAT activation by IFN-α [50]. Respiratory syncytial virus impairs IFN-β mediated signal transducer and activator of transcription *Stat1* phosphorylation through a mechanism that involves the inhibition of tyrosine kinase 2 phosphorylation [51,52,53]. The INF-α and INF-γ response pathways were highly activated in nearly all macrophage phenotypes with the treatment containing the caerin peptides; the expressions of many associated genes were largely upregulated in TAMs (Figure 3 and previously reported [29]). These two pathways were also remarkably activated in stage I cervical cancer patients compared to those in stage II (Figure 6). The most activated signalling pathway of stage II tumour macrophages is the TNF-α pathway, this accords with a recent study showing the serum level of TNF-α expression in patients with cervical cancer was noticeably elevated, which gradually became normal after the surgical treatment [54].

IFN-α has been used to treat cancers; however, the systemic administration of type I IFNs leads to significant side effects for patients [55]. Recently, the ER-associated molecule STING has been found to stimulate the production of type I IFN in the tumour [56,57] and results in anti-tumour responses through the activation of IFN-α signalling pathways in macrophages [58,59]. Moreover, targeting the IFN-α signalling pathway augments immune checkpoint inhibitors in ICB therapy-resistant tumours [60,61]. Interestingly and surprisingly, caerin 1.1/1.9 enhanced the anti-tumour responses of ICB and therapeutic vaccination therapy [23], also through activating the IFN-Stat1 signalling pathway of macrophages in TC-1 model (Figure 3 and Appendix A), and in the B16 tumour model. A recent study has shown that IFN-α induces the differentiation and exhaustion of chronic myeloid leukemia and myeloproliferative neoplasm stem cells via the processes mediated by *Cebpb*, and IFN-α upregulates C/EBPβ by recruiting *Stat1* and *Stat5* to the novel 3′ distal enhancers of *Cebpb* [62]. The upregulated expression of *Cebpb* was identified with significance in most MΦ phenotypes of the topical treatment containing the caerin peptides, which was also present as one of the differential TFs in the MΦs of the caerin + DT. The significant upregulation of *Stat1* was confirmed by qPCR in this study and was previously detected in both the topical application and intratumoural injection treatments containing the caerin peptides [23,29]. It was demonstrated that C/EBPβ promotes immunity to mucosal candidiasis during cortisone immunosuppression in a manner associated with the expression of β-defensin 3 [63]. Additionally, C/EBPβ regulates numerous genes involved in inflammation [64]. *Cebpb* and *Gm8797* (ubiquitin B pseudogene) were detected as the marker genes of the MΦs of *Vsir* knockout mice, which showed an exacerbated inflammatory phenotype [65]. The expression of *Gm8797* was remarkably upregulated in the caerin + DT. This further suggested *Cebpb* may play an important role in the proinflammatory state induced by caerin 1.1/1.9.

The depletion of macrophages or avoidance of macrophage tumour infiltration in the TC-1 tumour model results in either a beneficial or detrimental anti-tumour response. The depletion of myeloid cells by injecting clodronate liposome increased the efficacy of the vaccination, in terms of reducing tumour size and prolonging the survival of tumour-bearing mice [39]. A recent study showed that the vaccination with a long peptide plus incomplete Freund’s adjuvant-induced tumour regression, which was abrogated rather than enhanced by macrophage depletion [66]. These results indicated that whether macrophages promote tumour rejection or growth depends on their biological characteristics during cancer therapy. Compared with a control peptide, caerin 1.1/1.9 significantly activated tumour infiltrating macrophages, both F4/80+Ly6C+ or F/40+Ly6C− macrophages secreted significantly more IL12, less IL10 and IL6 (Figure 1), coinciding with our previous results that caerin 1.1/1.9 repolarise the tumour macrophage to the M1-like type while reducing the M2-like macrophages. Therefore, the caerin treatment resulted in macrophages becoming more proinflammatory, it was thus not surprising that the depletion of macrophages in our model attenuated the antitumor effect of therapeutic vaccination and ICB treatment. The depletion of tumour macrophages greatly reduced the enhanced anti-tumour responses of immunotherapy mediated by caerin 1.1/1.9 (Figure 2 and Appendix A). Our results and others point to the importance of activated macrophage type I IFN signalling pathways for the outcome of better immunotherapy. The induction of pro-inflammatory cytokines might result in the clonal selection of tumor cells and possible disease relapse in the context of caerin treatment; however, at least in the TC-1 model, the proinflammatory macrophages contribute to the tumor regression. To the best of our knowledge, this is the first time to demonstrate that natural host-defence peptides derived from amphibian skin secretion stimulate type I IFN signalling pathways in tumour MΦs. However, the molecular mechanism underlying this phenomenon is yet to be clear, it is likely that macrophages were activated indirectly, and that caerin peptides result in TC-1 tumour cell death, which release biomolecules that subsequently activate macrophages. We are currently investigating the underlying mechanism by using two murine tumour models and seeking to identify the target molecules of caerin 1.1/1.9.

Macrophage infiltration of some solid cancers commonly correlates with poor prognosis [67]. Targeting macrophages has long been used in different tumour models for cancer therapy with varied outcomes [37,39,68,69]. The regulation of macrophage polarisation via stimulating macrophage proinflammatory gene expression has been used to tackle cancer immunosuppression, and consequently activate cytotoxic T cell antitumor responses [70]. It has been reported that the expression of *Fos*/*Jun* enhances inflammatory responses in macrophages [71,72]. *Jund* was previously shown to be positively associated with inflammation, the knockdown of which resulted in significantly reduced macrophage activity and cytokine secretion in rat and human primary macrophages [73]; whereas *Junb* was found to play the key role in the full expression of *Il1b* and the genes involved in classical inflammation in macrophages treated with LPS and other immunostimulatory molecules [74]. A very recent study has discovered that the overexpression of *Fosl2* induced a systemic inflammatory phenotype with immune infiltrates in multiple organs in mice, by repressing the development of regulatory T cells [75]. These TFs were significantly elevated in the DT + caerin therapy compared to the untreated and control + DT groups, indicating more inflammatory phenotypes of MΦs formed in the TME. It is not surprising that several TFs associated with anti-inflammatory function were also elevated in the DT + caerin group, such as *Usf1*, *Nfil3*, and *Mafb* (Figure 4 and Appendix A).

On the other hand, several TFs negatively associated with inflammation were downregulated in the caerin + DT group compared to the DT treatments, such as *Ets2*, *Hif1a*, *Mitf*, *Runx1*, and *Elk3*. The negative regulatory role of *Ets2* in LPS- and VSV-induced inflammation through the suppression of MAPK/NF-κB signalling was revealed, via the inhibition of transcription of IL-6 with *Ets2* direct binding to the promoter [76]. The *Mitf*^−^ cells were shown to produce larger amounts of IL-1α and IL-1β, compared to *Mitf*^+^ cells; in addition, the supernatant of *Mitf*^-^ melanoma cells reduced *Mitf* expression in positive cells via the signalling of IL-1R, which we previously found was significantly upregulated in the caerin group [23]. The suppression of *Mitf* in melanoma cells triggers an inflammatory secretome comprising the proinflammatory cytokines [77,78]. It has been reported that the expression of *Runx1* is negatively associated with the production of inflammatory cytokine production by neutrophils in response to toll-like receptor signalling [79]. Besides, the downregulation of *Elk3* and the downstream induction of *Ho1* appeared crucial for the inflammatory response of macrophage function to infection [80]. Thus, the suppression of these TFs by the caerin + DT further supports the formation of a remarkably more proinflammatory TME.

Future experimentation with a higher number of CC patients at each stage is recommended to confirm the findings of this study and elucidate the underlying mechanism of the formation of distinct phenotypes. This may be achieved by a large-scale comparative transcriptomic analysis, in conjunction with the silencing of marker genes in murine models to verify the pathway(s) involved. At the protein level, antibody-mediated signalling blockade or activation by agonists could be performed, with a particular focus on macrophage phenotypes. With respect to how caerin 1.1/1.9 activate IFN signalling of macrophages, the knockout of key regulators on the pathway, e.g., *Stat1*, *Stat2*, *Jak1*, and *Ifnar*, as well as the use of condition knockout mice, could be employed, followed by in-depth multi-omic analysis. Besides, it should also be noted that angiogenesis plays a significant role in tumor progression, and its constant crosstalk with immune cells, including macrophages [81,82]. It has been shown that anti-angiogenic treatments could induce the normalisation of the tumor vasculature [83], which enhances anti-tumor immunity by increasing antigen presentation in dendritic cells, the populations of M1-like macrophages and active CD8+ T-cells [84]. These were also observed in the treatments containing caerin 1.1/1.9. Thus, caerin 1.1/1.9 in combination with anti-angiogenesis and immune checkpoint inhibitors in the treatment of different cancers warrant further investigation.

## 5. Conclusions

Taken together, for the first time, we unveil that caerin 1.1/1.9 are able to alter the tumour immune suppressive environment fundamentally to a more immune-active state, mainly through activating the type I type IFN signalling pathway of tumour MΦs, which significantly enhances the efficacy of ICB and warrants further investigation in more therapeutic vaccine-based immunotherapy.

## Figures and Tables

**Figure 1 cancers-14-05785-f001:**
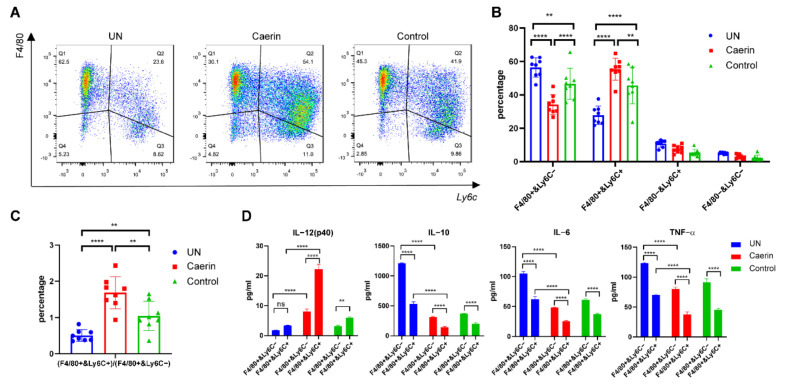
Mice receiving vaccination and ICB (Double Therapy, DT) immunotherapy plus caerin 1.1/1.9 have fewer F4/80+Ly6C− and more F4/80+Ly6C+ macrophages. After three days of the TC-1 challenge, the mice were randomly assigned when the tumours were palpable. The mice of experimental group were subjected to intratumoural injection of caerin 1.1/1.9 (Caerin) or a control peptide P3 (Control) from days 3 to 9. Additionally, they were immunised intramuscularly (i.m.) with the vaccine on day 3, day 9 and 18, respectively; 300 µg of anti-PD-1 was administered intraperitoneally (i.p.) on days 9 and 21 after the tumour challenge and the control was administered with PBS only. The mice were sacrificed on day 24 and the tumours were dissected for flow cytometry analysis. (**A**) Flow cytometry plots are shown for F4/80 and Ly6c staining of tumour-infiltrating myeloid cells, and four different subpopulations were distinguished. (**B**) The percentages of the four different subpopulations following therapy. The difference was statistically significant (ns: non-significant, **: *p* < 0.01, and ****: *p* < 0.0001). (**C**) The ratio of F4/80+Ly6C+ to F4/80+Ly6C− macrophages was highest following immunotherapy with Caerin, intermediate with immunotherapy with Control and lowest with PBS. (**D**) F4/80+Ly6C−, F4/80+Ly6C+ macrophages were sorted and stimulated with MPLA (100 ng/mL) overnight, and the levels of IL-12p40, IL-10, IL-6 and TNF-α derived from the supernatants of different treatment groups were measured by cytokine ELISA. The results shown are from one of two independent experiments.

**Figure 2 cancers-14-05785-f002:**
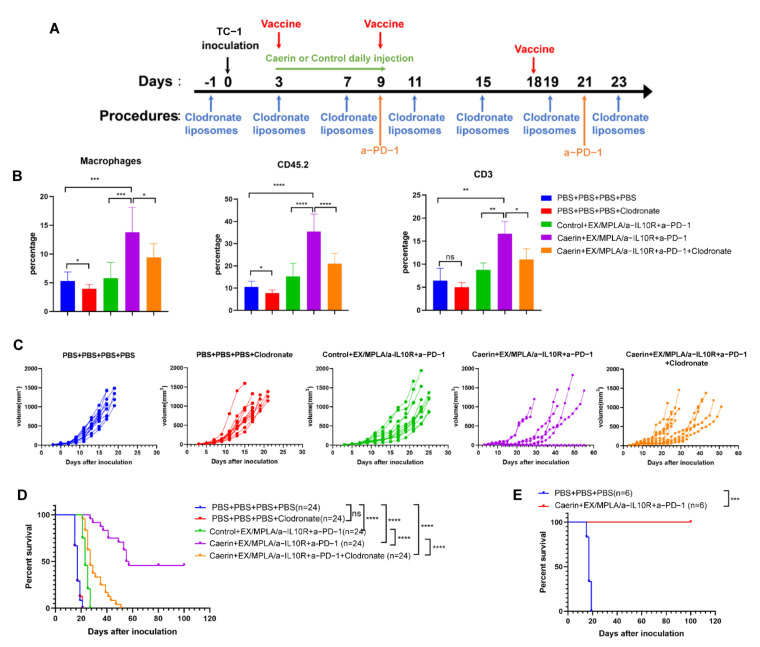
Depletion of macrophages impacts DT with caerin 1.1/1.9. (**A**) Timeline of immunotherapy and macrophage depletion. (**B**) Tumour infiltrating macrophages (CD45+CD11b+F4/80+), CD45+ and CD3+ T cells for the different treatment groups. (**C**) Tumour volume growth curves by treatment; each curve represents one individual. (**D**) Survival curves in different groups (*n* = 24). (ns: non-significant, *: *p* < 0.05, **: *p* < 0.01, ***: *p* < 0.001 and ****: *p* < 0.0001). (**E**) Survival curves for mice that cleared the first TC-1 inoculation after triple therapy and rechallenged with TC-1 tumour (*n* = 6).

**Figure 3 cancers-14-05785-f003:**
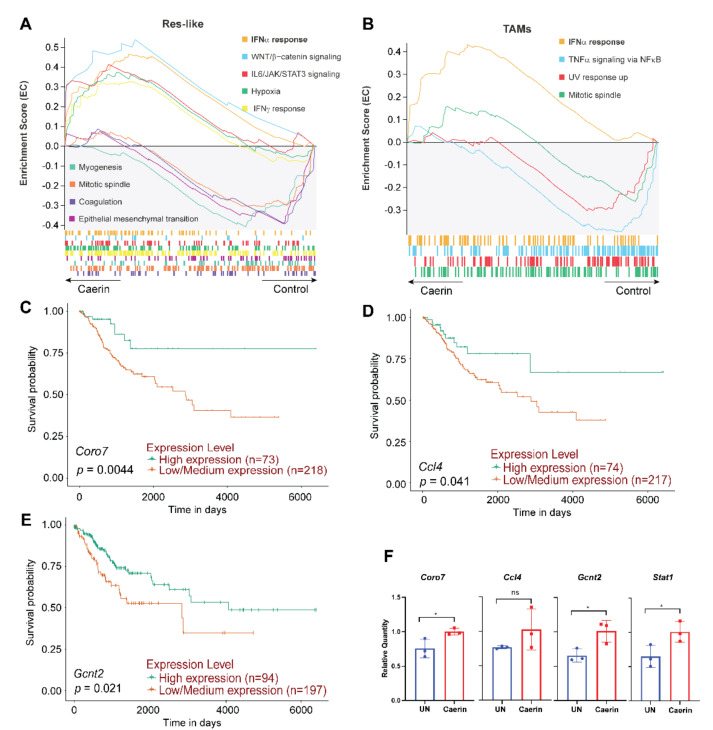
Macrophage phenotypes are regulated by the topical application of the gel containing caerin 1.1/1.9. Enrichment of signalling pathways in resident-like macrophage (**A**) and TAMs (**B**) of the caerin group relative to the control group predicted by GSEA analysis of the data published previously [29]. The correlation between survival curves of cervical squamous cell carcinoma and the expression of selected genes identified in (Appendix A) derived from the TCGA data portal, including *Coro7* (**C**), *Ccl4* (**D**), and *Gcnt2* (**E**). (**F**) Comparison of the expression of *Coro7*, *Ccl4*, *Gcnt2*, and *Stat1*, using qPCR (*n* = 3). Student’s t-test was used to assess the significance, *: *p* < 0.05, ns: non-significant.

**Figure 4 cancers-14-05785-f004:**
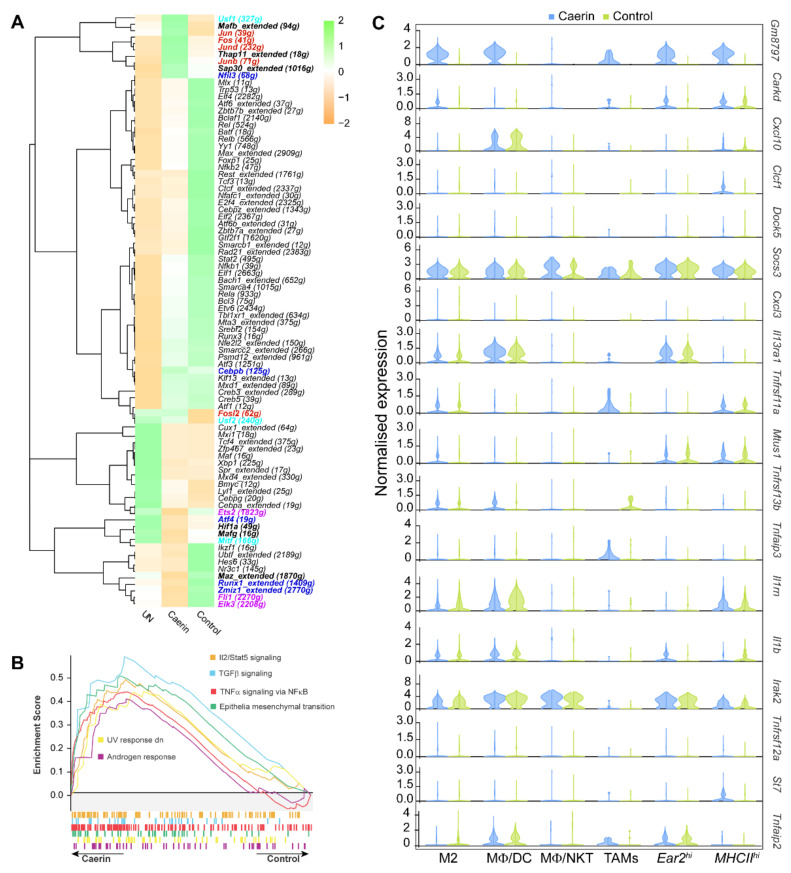
Regulation of MΦ populations following DT therapy with or without caerin. (**A**) SCENIC analysis of the significant transcription factors following different treatments; the bold font labels significantly regulated transcription factors in the caerin group. The analysis was performed based on the scRNA-seq data published previously [23]. (**B**) GSEA analysis of the Hallmark pathways enriched in the *Ear2^hi^* MΦs of the caerin group, relative to the control. (**C**) The top 18 immune response-relevant genes significantly regulated by treatment containing the caerin peptides.

**Figure 5 cancers-14-05785-f005:**
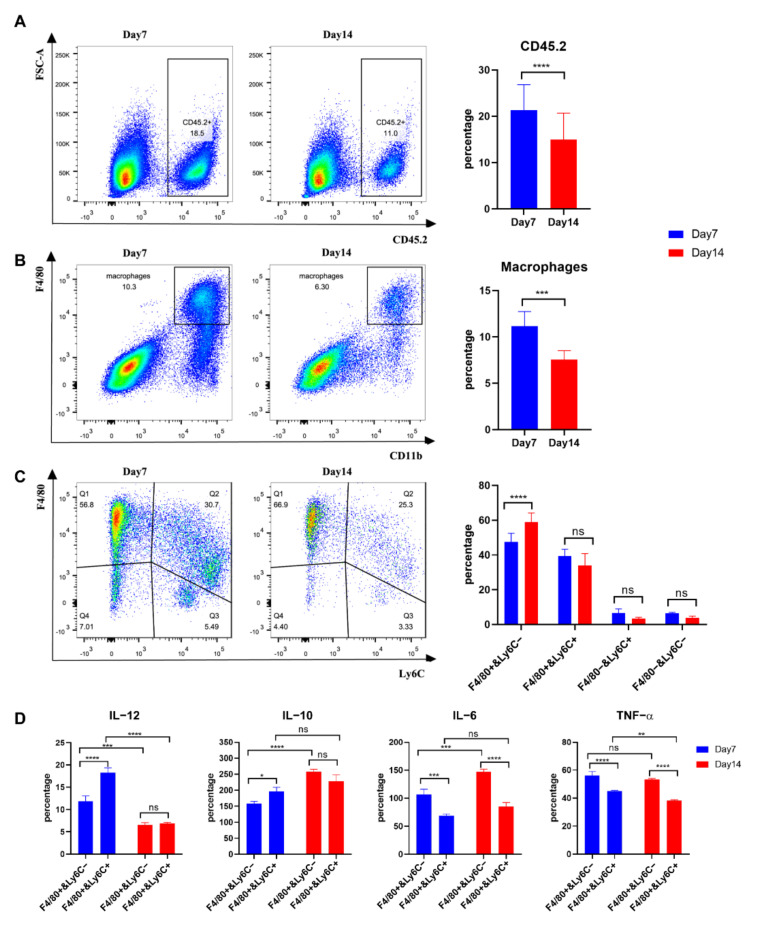
Comparison of the macrophage phenotypes in 7- and 14-day TC-1 tumours. (**A**) Comparison of CD45+ cell populations; (**B**) Comparison of overall macrophage populations; (**C**) comparison of four subpopulations of macrophages, including F4/80+Ly6C^−^, F4/80+Ly6C+, F4/80-Ly6C−, and F4/80-Ly6C+; (**D**) secretion of IL-12, IL-10, IL-6 and TNF-α by different MΦ subpopulations at early- or late-stage. Student’s *t*-test was used to evaluate the significance (ns: non-significant, *: *p* < 0.05, **: *p* < 0.01, ***: *p* < 0.001 and ****: *p* < 0.0001).

**Figure 6 cancers-14-05785-f006:**
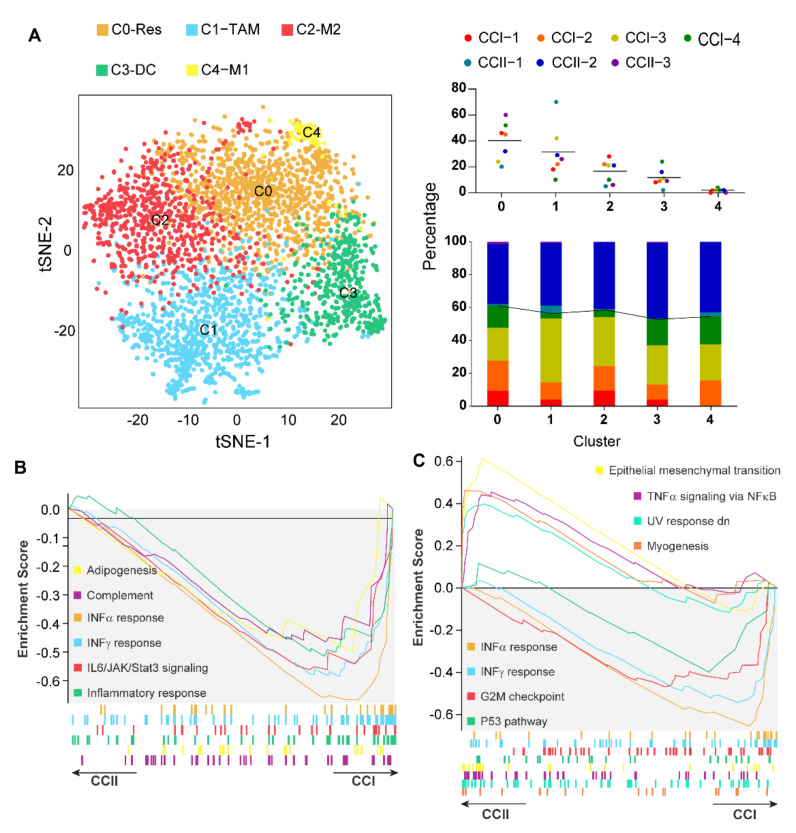
Comparative analysis of the functions of macrophages in the TME of early (CCI) and late (CCII) stage cervical cancer patients. (**A**) 2D t-SNE distribution of macrophage subtypes identified, including resident-like, TAM, M2-like, MΦ/DC, and M1-like MΦs (**left**). Proportions of each MΦ subpopulation in the MΦs of the patients (**top right**) and the proportion of individual patient MΦ in each subpopulation (**bottom right**). GSEA analysis of the enriched Hallmark pathways in M1-like MΦs (**B**) and TAMs (**C**).

**Table 1 cancers-14-05785-t001:** Real time PCR primer pairs of selected genes.

Gene	Primer Direction	Sequence (5′–3′)	Tm (°C)
*Stat1*	Forward primer	ACCACCTCTCTTCCTGTCGT	55.6
	Reverse primer	GATTCCTGGGCTCTGTCACC	59
*Ccl4*	Forward primer	CCTTCTGTGCTCCAGGGTTC	59.3
	Reverse primer	TGCCTCTTTTGGTCAGGAATAC	58.7
*Cav1*	Forward primer	GCAGACGAGGTGACTGAGAA	55.5
	Reverse primer	AAATGCCCCAGATGAGTGCC	62
*Coro7*	Forward primer	CAGCAGCCCTTTACCCCTAC	59.1
	Reverse primer	CACCATCACACACCCAGACA	57.2
*Gnt2*	Forward primer	GCAATGGGCGACAAAGGAA	61.6
	Reverse primer	TGCGAAGCGAGTCTATCAGC	59
*Actb*	Forward primer	GCTGTCCCTGTATGCCTCTG	57.8
	Reverse primer	ATGTCACGCACGATTTCCCT	60.1

**Table 2 cancers-14-05785-t002:** The list of genes differentially expressed between the MΦs of the CCI and CCII groups, that were also significantly regulated in the MΦs of the caerin groups of topical gel and triple therapy treatments [23,29].

Avg. Exp	Patient			Topical Gel			Triple Therapy (DT + Control/Caerin)		
	CCI	CCII	MΦ Type	Control	Caerin	MΦ Type	Control	Caerin	MΦ Type
*Fn1*	3.958464	0.756023	MΦ/DC	8.651744	6.683293	Arg1B	1.509254	0.796691	M2
	1.104985	3.562443	M2						
*Vegfa*	2.032601	3.077626	M2	1.381038	1.730747	MΦ/DC	0.436143	0.254142	MΦ/DC
	1.530725	2.29818	TAM						
	0.781789	2.755229	MΦ/DC						
	0.159604	0.844889	M1						
*Aig1*	1.03329	0.633123	M2	0.151279	0.272457	Resident	0.371028	0.246382	Ear2^hi^
	1.174192	0.529877	MΦ/DC						
	1.376729	0.733273	TAM						
*Slamf7*	1.039648	0.477852	M2	0.931226	1.191878	Arg1A	2.951016	1.864526	MHCII^hi^
	0.752496	0.360924	Resident						
*Igsf6*	0.899768	0.273307	MΦ/DC	0.5863	0.481483	M1	0.439447	1.820253	MΦ/NKT
				0.889448	0.69455	Arg1A			
*Rnf130*	4.960039	3.241792	TAM	0.492294	0.612842	M1	0.632459	1.614695	MΦ/NKT
*Dock5*	2.297257	3.45598	TAM	0.183134	0.150017	Arg1B	0.11796	0.218823	MΦ/DC
*Rilpl2*	0.712195	0.269815	MΦ/DC	0.51496	0.389841	Resident	0.424038	0.229098	MΦ/DC
				0.408273	0.320012	MΦ/DC			
*Pim3*	0.414601	0.833308	MΦ/DC	0.499359	0.838651	Ear2^hi^	0.194824	0.325949	MΦ/DC
	0.40287	0.641915	Resident						
	0.250678	1.079287	TAM						
*Spint1*	0.301725	0.978266	TAM	1.392293	1.701847	Arg1B	0.832327	1.36062	Ear2^hi^
				0.43993	0.53739	Arg1A			
*Cd151*	0.240536	0.750399	TAM	0.475691	0.340939	Arg1A	0.360751	0.55949	MΦ/DC
	0.200822	0.594903	M2						
	0.182115	0.54708	MΦ/DC						
	0.160279	0.403624	Resident						
*Hivep2*	1.929148	3.051844	MΦ/DC	0.103692	0.182033	M1	0.16374	0.106456	M2
*Tnfaip3*	1.254977	2.087381	MΦ/DC	0.574416	0.315302	Arg1A	0.151357	0.334904	Ear2^hi^

## Data Availability

Datasets of single-cell RNA sequencing related to the intratumoural injection and topical application treatments containing caerin 1.1/1.9 on murine models are hosted at the Institute Single Cell Portal and can be found at https://singlecell.broadinstitute.org/single_cell (accessed on 13 May 2022), the accession number SCP1371 and SCP1093. Datasets of single-cell RNA sequencing related to CC patients can be found at (https://singlecell.broadinstitute.org/single_cell, the accession number SCP1950). The datasets of flow cytometry can be found at the Flowrespository [85] (https://flowrepository.org/id/RvFrThLokoMtf5zUbvQuy3GqiGDqu8CT2O6Mgp3ngUaZUSWfqhqv0REBxRQUqEKY (uploaded on 26 August 2022)).

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
