# Peer review of "Caerin 1.1/1.9 Enhances Antitumour Immunity by Activating the IFN-α Response Signalling Pathway of Tumour Macrophages"

_cancers, 2022, doi:10.3390/cancers14235785_

Round 1

Reviewer 1 Report

What about the effect of Caerin on other cellular compartments of the immune system, like T cells, and NK cells?

Induction of pro-inflammatory cytokines could also result in clonal selection of tumor cells and possible disease relapse, and more aggressive disease. Can the authors discuss that possibility in the context of caerin treatment, or in vitro co-culture assays?

Please correct the English language of the manuscript, taking help of a professional.

Author Response

Response to Reviewer 1 Comments

Point 1:What about the effect of Caerin on other cellular compartments of the immune system, like T cells, and NK cells?

Response 1: Our previous studies have demonstrated that caerin 1.1/1.9, whether intratumorally injected or transdermal delivered attract more NK and T cells to the tumour sites, and these cells were more activated compared with the treatments with a control peptide or untreated tumour (Ma et al., American Journal of translational research 2020; Ni et al., Frontiers in Oncology 2021; Ni et al., Clinical and Translational Immunology 2021). These have been included and highlighted in the Introduction, Page 2 Line 82 to 85.

Point 2: Induction of pro-inflammatory cytokines could also result in clonal selection of tumor cells and possible disease relapse, and more aggressive disease. Can the authors discuss that possibility in the context of caerin treatment, or in vitro co-culture assays?

Response 2: Thanks for the comments and suggestions, we have included more discussion on the induction of pro-inflammatory cytokines might result in clonal selection of tumor cells and possible disease relapse in the discussion in the context of caerin treatment. This content has been added on page 17, line 554 to 556 of the revised manuscript.

Point 3: Please correct the English language of the manuscript, taking help of a professional.

Response 3: The revised manuscript has been thoroughly edited by a professional native English speaker. The changes have been highlighted.

Reviewer 2 Report

The manuscript represents continuation of  previous studies that demonstrated caerin antitumor effects, using a TC-1 murine tumor model for HPV-associated cancers. Here, using the same animal model, the potential of caerin1.1/1.9 to enhance antitumor combined immunotherapy was evaluated and a special attention was paid to the role of tumor-associated macrophages (TAMs). The authors concluded that IL-12 secreting Ly6C+F4/80+ macrophages with activated  IFN alpha signaling pathway are crucial  mediators of the caerin effects. Importantly, macrophages elimination by clodronate abolished the caerin therapeutic effects. The animal study has been completed with the analysis of a limited number of  human carcinoma samples documenting that IFN-alpha response signaling pathways in activated macrophages were increased rather in the early than late stages of the disease. The study brings some new data on mechanisms how TAMs modulation can influence antitumor immune responses, as well as they support caerin immunotherapeutic potential. However, the study suffers from some shortcomings and several points should be clarified or completed. Indeed, the mechanisms by which caerin can activate IFN signaling in TAMs has not been investigated, neither discussed in detail.

Specific comments:

1.       In Fig. 1 it is not clear at what time points the samples were collected. Was it at the end of the experiment i.e. when the tumors reached 15 mm diameters?

2.       It is not clear whether the caerin effects on macrophages is direct or not. An experiment in vitro confirming or excluding the directs effects (e.g. determination of the STAT1 phosphorylation) should be performed.

3.       Is there anything known whether and by which mechanisms could caerin (directly ?) activate IFN pathways in immune or tumor cells?

4.       Recently, direct effects of Caerin on tumor cells have been described in  NSCLC tumor cells (Liu n et al., Frontiers in Oncology, 2022). Have you checked the direct effects on the TC-1 cells? This point should be discussed or verified.

5.       The number of the patients’ samples for analysis was quite low. Can you discuss the statistical significance of your conclusions?

Author Response

Response to Reviewer 2 Comments

The manuscript represents continuation of previous studies that demonstrated caerin antitumor effects, using a TC-1 murine tumor model for HPV-associated cancers. Here, using the same animal model, the potential of caerin1.1/1.9 to enhance antitumor combined immunotherapy was evaluated and a special attention was paid to the role of tumor-associated macrophages (TAMs). The authors concluded that IL-12 secreting Ly6C+F4/80+ macrophages with activated IFN alpha signaling pathway are crucial mediators of the caerin effects. Importantly, macrophages elimination by clodronate abolished the caerin therapeutic effects. The animal study has been completed with the analysis of a limited number of human carcinoma samples documenting that IFN-alpha response signaling pathways in activated macrophages were increased rather in the early than late stages of the disease. The study brings some new data on mechanisms how TAMs modulation can influence antitumor immune responses, as well as they support caerin immunotherapeutic potential. However, the study suffers from some shortcomings and several points should be clarified or completed. Indeed, the mechanisms by which caerin can activate IFN signaling in TAMs has not been investigated, neither discussed in detail.

Response: We sincere thank the reviewer for the comments of our manuscript. In the current study, the enhanced efficacy of immunotherapy mediated by tumour local treatment of caerin peptides were mediated by macrophages, most likely through IFNα response signalling pathway as demonstrated by the scRNA-seq analysis of the tumour infiltrating macrophages. Our scRNA-seq analysis of macrophages between human stage-I and II cervical cancer tumor samples also indicate that the IFNα response signalling pathway was more activated in the stage-I cervical cancer, which normally shows a higher and more active immune response to tumorigenesis with respect to other stages. This is the first time, to the best of our knowledge, that natural host-defence peptides identified from amphibian skin secretions are able to modulate the tumour microenvironment, via reprogramming the phenotypes of tumour infiltrating macrophages, from tumor promoting M2-like to M1-like type, leading to improved efficacy of immunotherapy. Indeed, the molecular mechanism underlying this phenomenon is yet to be clear, we are working on it by using two murine tumor models. The significance of caerin mediated improvement of immunotherapy have been now added in the revised manuscript, which is on page 16, line 481 to 489.  

Specific comments:

Point 1: In Fig. 1 it is not clear at what time points the samples were collected. Was it at the end of the experiment i.e. when the tumors reached 15 mm diameters?

Response 1: The date that the tumors were dissected were added to the revised manuscript as below:

“After three days of the TC-1 challenge, the mice were randomly assigned when the tumors were palpable. The mice of experimental group were subjected to intratumoral injection of caerin 1.1/1.9 or a control peptide P3 from days 3 to 9. Additionally, they were im-munised intramuscularly (i.m.) with the vaccine on day 3, day 9 and 18 respectively. 300 µg of anti-PD-1 was administered intraperitoneally (i.p.) on days 9 and 21 after tumour challenge and the control group was administered with PBS only. The mice were sacrificed on day 24 and the tumors were dissected for flow cytometry analysis.”

These were added in the revised manuscript on page 7, line 274 to 279.

Point 2: It is not clear whether the caerin effects on macrophages is direct or not. An experiment in vitro confirming or excluding the directs effects (e.g. determination of the STAT1 phosphorylation) should be performed.

Response 2: Thanks for this comment, this is similar to question 3, it will be addressed together in the response to question 3.

Point 3: Is there anything known whether and by which mechanisms could caerin (directly ?) activate IFN pathways in immune or tumor cells?

Response 3: It is unclear whether caerin peptides activate macrophage IFNα signaling pathways directly or indirectly, we are currently working on this important issue, it is most likely that macrophages were activated indirectly. Caerin peptides result in TC-1 tumor cell death, which will release cytokines and DNA or RNA that will subsequently activate macrophages. These has been added in the revised manuscript on page 17, line 559 to 563.  

Point 4: Recently, direct effects of Caerin on tumor cells have been described in NSCLC tumor cells (Liu n et al., Frontiers in Oncology, 2022). Have you checked the direct effects on the TC-1 cells? This point should be discussed or verified.

Response 4: Actually, several of us are the co-authors of the paper mentioned by the reviewer. Yes, we had published the data demonstrating the direct effects of caerin on TC-1 cells (Ni et al, Biomedical Research International, 2018; Pan et al, BMC complementary and alternative medicine, 2019). We demonstrated that caerin 1.1/1.9 inhibit TC-1 growth both in vivo either through intratumoral injection or through transdermal delivery in the form of temperature sensitive gel (Ni et al, Frontiers in Oncology 2021; Ni et al, Clinical and translational immunology, 2021). These contents and the associated references have been added in the revised manuscript on page 2, line 79 to 81.

Point 5: The number of the patients’ samples for analysis was quite low. Can you discuss the statistical significance of your conclusions?

Response 5: We agree with the reviewer that a higher number of patients might help draw more conclusions. However, the current study has used the tumour tissue samples from seven patients (incl, four stage-I and three stage-II) in the scRNA-seq analysis, which is a reasonable number to reveal the most significant differences in the heterogeneity and functions of macrophages in the tumour microenvironment. The scRNA-seq data were filtered under strict QC control to obtain the cells with most confident expression data. Also, the macrophages are the cell types with relatively high cell numbers in the CCI and CCII, meaning the impact of replicate numbers to the statistically significant results should be low, which supports the confidence in the findings.  

Reviewer 3 Report

The authors uncovered interesting topic regarding immunobiology and macrophages.

Points to be addressed:

1) The rationale of why the authors came up with this research is scanty and is related to a lack of novelty: please highlight what this manuscript might add.

2) What is the information that is not exactly available that motivated the authors to come up with this information. What are the current caveats and how do the authors highlight the current research in answering them? If not they need to address in background and infuture directions .

3)State of the art figures are required: scale bar should be provided in high resolution.

4)The authors could provide a little more consideration of genomic directed stratifications in clinical trial design and enrolments. 

5)The underlying message here is that more precision and individualized approaches need to be tested in well-designed clinical trials – a challenge, but I would be interested in their perspective of how this might be done. If beyond the scope of the manuscript, this should be highlighted as a limitation

6) The authors need to highlight what new information the review is providing to enhance the research in progress

7) did the author employ unstained/isotype controls?

8)this reviewer personally misses some insights regarding cancer angiogenesis, macrophages and tumor immunity: as is now well known, tumors grow and evolve through a constant crosstalk with the surrounding microenvironment, and emerging evidence indicates that angiogenesis and immunosuppression frequently occur simultaneously in response to this crosstalk. Accordingly, strategies combining anti-angiogenic therapy and immunotherapy seem to have the potential to tip the balance of the tumor microenvironment and improve treatment response.Resistance to anti-vascular endothelial growth factor (VEGF) molecules causes lack of response and disease recurrence. Acquired resistance develops as a result of genetic/epigenetic changes conferring to the cancer cells a drug resistant phenotype. In addition to tumor cells, tumor endothelial cells also undergo epigenetic modifications involved in resistance to anti-angiogenic therapies. The association of multiple anti-angiogenic molecules or a combination of anti-angiogenic drugs with other treatment regimens have been indicated as alternative therapeutic strategies to overcome resistance to anti-angiogenic therapies. Alternative mechanisms of tumor vasculature, including intussusceptive microvascular growth (IMG), vasculogenic mimicry, and vascular co-option, are involved in resistance to anti-angiogenic therapies. The crosstalk between angiogenesis and immune cells explains the efficacy of combining anti-angiogenic drugs with immune check-point inhibitors. Collectively, in order to increase clinical benefits and overcome resistance to anti-angiogenesis therapies, pan-omics profiling is key (please refer to PMID: 34298648 and expand)

Author Response

Response to Reviewer 3 Comments

The authors uncovered interesting topic regarding immunobiology and macrophages.

Point 1: The rationale of why the authors came up with this research is scanty and is related to a lack of novelty: please highlight what this manuscript might add.

Response 1: The rational of current research aligns with our several previous studies, we found that caerin peptides were able to inhibit the growth of TC-1 and HeLa cells in vitro, which are all HPV+ cell lines, and induce cell apoptosis. In vivo, we discovered that caerin 1.1/1.9 increases the efficacy of ICB and therapeutic vaccine mediated immunotherapy on murine TC-1 models, either with intratumoral injection or topical application. We then comparatively investigated the cell heterogeneity and functions in the TME of TC-1 tumours in murine models with the caerin 1.1/1.9 treatments, which revealed that the phenotypes of macrophages were reprogrammed by caerin 1.1/1.9. Meanwhile, we have recently compared the cell types and functions of stage-I and II human cervical cancer patients (also HPV+) using scRNA-seq in conjunction with deep tissue proteomic analysis, which showed that there are high similarities in macrophage phenotypes between stage-I CC and TC-1 tumour treated by caerin 1.1/1.9. This inspired us to revisit our previous scRNA-seq data, and thus designed experiments to verify that the macrophages play key role in anticancer activity of caerin 1.1/1.9 in current study. We also seek to further understand the underlying mechanism in future studies.

Point 2: What is the information that is not exactly available that motivated the authors to come up with this information. What are the current caveats and how do the authors highlight the current research in answering them? If not they need to address in background and in future directions.

Response 2: Immunotherapy has become a routine modality for the treatment of cancers, especially with the introduction of ICB and CAR-T therapy, therapeutic vaccines in pre-clinical animal models have also demonstrated exciting efficacies. However, a great portion of cancer patients do not respond to ICB treatment. One of the most important caveats is that the tumor microenvironment (TME), which is immunosuppressive, stinted the therapeutic effects of immunotherapy. Disturbing the TME is therefore a key issue to increase the efficacy of immunotherapy. Given caerin 1.1/1.9 can directly kill tumor cells and make the tumor cells release inflammatory cytokines, we therefore set to investigate whether caerin 1.1/1.9 can improve the TME and the efficacy of immunotherapy, and how.

The enhanced efficacy of immunotherapy mediated by tumor local treatment of caerin peptides were mediated by macrophage, most likely through IFNα signaling pathway as demonstrated by the single cell RNA sequencing analysis of the tumor infiltrating macrophages. scRNA sequencing analysis of macrophages between early and later stage cervical cancer tumor samples also indicate that the IFNα signaling pathway was more activated in the early-stage cervical cancer. This is the first time, to the best of our knowledge, that natural peptides isolated from amphibian skin secretions are able to modulate the tumour microenvironment, through improving the function of tumor infiltrating macrophages, from tumor promoting M2 type, to M1 type, which resulted in improved efficacy of immunotherapy.

These statements are now added in the revised manuscript on page 16, line 481 to 489.

Point 3: State of the art figures are required: scale bar should be provided in high resolution.

Response 3: The figures provided are of high quality that meet the standard of the journal. Our figures do not need any scale bar.

Point 4: The authors could provide a little more consideration of genomic directed stratifications in clinical trial design and enrolments. 

Response 4: This question is similar to next question, we will answer this question together with question 5.

Point 5: The underlying message here is that more precision and individualized approaches need to be tested in well-designed clinical trials – a challenge, but I would be interested in their perspective of how this might be done. If beyond the scope of the manuscript, this should be highlighted as a limitation.

Response 5: Thanks for the excellent suggestion. It will greatly enhance the quality of the manuscript if more genomic directed classification, more patients were recruited and better designed clinical investigation is to be executed. However, as pointed by the reviewer, this is beyond the scope of the manuscript. The aim of current paper is to investigate whether tumour infiltrating macrophage is responsible for the increased efficacy of immunotherapy induced by caerin 1.1/1.9. The scRNA-seq results indicate caerin treatment resulted in the activation of macrophage IFNα signaling pathway, while the activation of macrophage IFNα signalling pathway was also observed in early but not late-stage cervical cancer patients. Indeed, the results from cervical cancer patients need to be further investigated to confirm the above results, ideally with a higher number of patients at each stage. Limitations of this study was discussed in the revised manuscript on page 18, line 597 to 606.    

Point 6: The authors need to highlight what new information the review is providing to enhance the research in progress.

Response 6: We have revisited the data of two previous studies, to reveal the reprogramming of macrophage phenotypes of TC-1 tumour treated with caerin 1.1/1.9. In addition, we carried out significant amount of in vitro and in vivo assays, to testify that macrophages play a key role in the bioactivity of caerin 1.1/1.9. We have added more discussion to address the research could be enhanced, page 18, line 597 to 609.

Point 7: Did the author employ unstained/isotype controls?

Response 7: Yes, unstained or isotype controls were used in flow cytometry.

Point 8: This reviewer personally misses some insights regarding cancer angiogenesis, macrophages and tumor immunity: as is now well known, tumors grow and evolve through a constant crosstalk with the surrounding microenvironment, and emerging evidence indicates that angiogenesis and immunosuppression frequently occur simultaneously in response to this crosstalk. Accordingly, strategies combining anti-angiogenic therapy and immunotherapy seem to have the potential to tip the balance of the tumor microenvironment and improve treatment response. Resistance to anti-vascular endothelial growth factor (VEGF) molecules causes lack of response and disease recurrence. Acquired resistance develops as a result of genetic/epigenetic changes conferring to the cancer cells a drug resistant phenotype. In addition to tumor cells, tumor endothelial cells also undergo epigenetic modifications involved in resistance to anti-angiogenic therapies. The association of multiple anti-angiogenic molecules or a combination of anti-angiogenic drugs with other treatment regimens have been indicated as alternative therapeutic strategies to overcome resistance to anti-angiogenic therapies. Alternative mechanisms of tumor vasculature, including intussusceptive microvascular growth (IMG), vasculogenic mimicry, and vascular co-option, are involved in resistance to anti-angiogenic therapies. The crosstalk between angiogenesis and immune cells explains the efficacy of combining anti-angiogenic drugs with immune check-point inhibitors. Collectively, in order to increase clinical benefits and overcome resistance to anti-angiogenesis therapies, pan-omics profiling is key (please refer to PMID: 34298648 and expand)

Response 8: Thanks for the comments, relevant information and the reference provided were now added in the revised manuscript on page 18, line 606-623. We agree with the reviewer that there should be a better control in using caerin 1.1/1.9 to alter the TME, maybe a combination with anti-angiogenic drugs is worth looking at. We shall investigate the interaction between caerin 1.1/1.9 and the endothelial cells, and angiogenesis in a future study.

Round 2

Reviewer 2 Report

The manuscript has been thoroughly revised and all important problems were addressed. In my opinion, the manuscript can now be  published as it stands.

Reviewer 3 Report

The authors have clarified several of the questions I raised in my previous review. Most of the major problems have been addressed by this revision.